# The impact of civil unrest on child health care: Evidenced by acute medical complications at presentation – A retrospective comparative study

Tesfaye Taye Gelaw[1]*, Mensur Azeze Getahun[2], Assefa Mitiku Bayih[2], Kassahun Gedefie Hailu[2], Gasha Amsalu Tadesse[2], Gebeyaw Lulie Adamu[2], Mastewal Addisu Legesse[2]

1 Department of Pediatrics and Child Health, College of Medicine and Health Sciences, Bahir Dar University, Bahir Dar, Ethiopia, 2 Tibebe-Ghion Specialized Teaching Hospital, Bahir Dar University, Bahir Dar, Ethiopia

* ttgela2020@gmail.com

## Abstract

### Background

Civil unrest is a collective term that includes limited political violence, sporadic violent collective action, or nonviolent and mildly violent collective action that causes dissatisfaction over political, economic, or social changes. It had deadly impacts on the lives of children and adolescents. It often results in difficulties for civilian to access basic services including healthcare.

### Objective

Evaluate the effect of civil unrest on child healthcare provision evidenced by the proportion of children admitted with acute medical complications.

### Methods

Institution-based retrospective comparative study of Difference in Difference with Propensity Score Matching (PSM-DID) was implemented.

### Setting and participants

The study was conducted on children admitted to pediatric ward of Bahir Dar University Tibebe-Ghion specialized teaching hospital. Data were collected from medical records for the months of January 01, 2023 – July 31, 2023 (pre-civil unrest) and August 01, 2023 – February 29, 2024 (into the civil unrest), on a retrospective basis. 632 Participants (345 in the treatment and 247 in the control group) were selected randomly using Microsoft Excel based on their medical record number (MRN) from the HMIS registry with treatment assignment (rural residency or not).

**Data availability statement:** All relevant data are within the article and its Supporting information files.

**Funding:** The author(s) received no specific funding for this work.

**Competing interests:** The authors have declared that no competing interests exist.

## Results

PSM was conducted on 7 covariates. In the unmatched sample, significant differences between groups were found for two of the 7 covariates. PSM successfully adjusted for bias in all covariates in the matched sample. The civil unrest has increased acute medical complications at presentation to our hospital for rural residents, with a DID value of 0.241 (p-value = 0.009).

## Conclusion

Our study has concluded that civil unrest has an immediate impact on child health care evidenced by an increased proportion of acute medical complications at presentation. It affects more children coming from rural areas compared to those from urban communities.

## Introduction

Child health and well-being are well documented in the set targets and indicators of the United Nations' Sustainable Development Goals [1]. Children's well-being is put as a plan in both high-income countries (HICs) and low-income countries (LMICs) as it is the most important endowment in human and social capital investment of the future of a country [2–4]. Child well-being embraces a broad concept including children's health, safety, education, socialization, and the love and values instilled in them by the societal environment and families [5]. Even though children's health is affected by multiple and highly intertwined factors and is difficult to single out, environmental factors like civil unrest have a significant effect on child healthcare provision [6,7].

Civil unrest is a collective term that includes limited political violence, sporadic violent collective actions, or nonviolent and mildly violent collective action that causes tension over political, economic or social changes [8]. Armed conflict, political demonstrations, riots, and other forms of civil unrest are often marked by high levels of violence and are sometimes responded to with military force [9]. It occurs most often when participants of civil disobedience become antagonistic towards the authorities, and authorities must struggle to stop the initiative from an unruly crowd [10].

Armed conflicts are frequently characterized by damage to health infrastructure, and injury or death of health workers which interrupt the health services delivery [11,12]. Healthcare providers are unfortunately not spared from violence in conflict times. This will affect service provision in the health facilities in conflict-hit areas by reducing the number and quality of healthcare providers because of loss of health care providers' life, fleeing to other area, another job opportunity, involvement in military services and food insecurity [13,14]. It has also a deadly impact on the lives of children and adolescents [15]. It often results in difficulties for the civilian population to access basic services such as healthcare and education [16]. In addition to dying from direct effects of the war, children die more from starvation, illness and complications or a combination of factors involving the loss of their parents during and in the aftermath periods [17]. Civil unrest increases vulnerability to infectious and non-infectious diseases and nutritional deprivation because of multiple environmental factors including lack of access to basic healthcare. In resource-limited countries, conflict affects significantly the already fragile health systems through health clinic shutdowns. Blockage of transport systems and violent attacks on healthcare facilities significantly affect civilian access to health services [18,19]. Hence, it is expected that children with illnesses, who otherwise, access health facilities early may present late with complications.

Access to health facilities is becoming one of the challenges for sick children to get timely optimal treatment because of the civil unrest and the associated multiple factors. As a result of limited access to health care services during civil unrest, improving and even maintaining child health care provision remains a significant challenge which in turn results in an increased number of children presenting to health facilities with acute medical complications. Additionally, access to health care services is dependent on availability, geography, accommodation, affordability, and acceptability; all of which are often undermined by the presence of conflicts [20–22]. Civil unrest become more common and more physically destructive in the last five years, with devastating consequences for health care provision in LMICs including our country [7]. Children in Northwest and North-central Ethiopia are facing sufferings from the civil unrest in the country.

So far, there is limited information regarding the impact of civil unrest on child healthcare provision in the region [15]. Having such a piece of information would provide very relevant information to further evaluate the impact of civil unrest on child healthcare provision. Cognizant of this, we aimed to assess the direct impact of civil unrest on children's health using acute medical complications at presentation as an outcome indicator in children admitted to our hospital. This will provide evidence-based insight to the community leaders, the healthcare industry, the government and the international community to focus on strengthening the health systems in shouldering such a burden in times of civil unrest.

## Methods

### Setting and participants

The study was conducted on children admitted to pediatric ward of Bahir Dar University Tibebe-Ghion specialized teaching hospital, a large teaching and referral center for pediatric patients from Northwest, North Central, and part of western Ethiopia with an estimated annual admission capacity of 1,500–2,000 children. Data were collected from medical records for the months of January 01, 2023 – July 31, 2023 (group pre-civil unrest period) and August 01, 2023 – February 29, 2024 (group into the civil unrest period), on a retrospective basis from the archives of the Hospital between June 08, 2024, and June 17, 2024. Data were collected by trained nurses and residents retrospectively using structured questionnaires from medical records of children admitted to the pediatric ward.

### Study design

Institution-based retrospective comparative study of Difference-in-Differences with Propensity Score Matching (PSM-DID) was implemented. Patients' medical files were selected randomly using Microsoft Excel based on their medical record number (MRN) taken from HMIS registries in the respective wards and data on socio-demographic parameters, anthropometry measurements, vaccination status, residency, approximate distance from the hospital, type of medical illnesses, and presence of acute medical complications at presentation to our hospital were retrieved from the medical records of clients seen pre-civil unrest (January 01, 2023 – July 31, 2023) and into the civil unrest (August 01, 2023 – February 29, 2024) periods retrospectively. This study was approved by the Institutional Review Board of Bahir Dar University College of Medicine and Health Sciences without comments (protocol number: 1044/2024).

### Sample size and selection

Sample Size formula for two independent samples with dichotomous outcomes was used to test our null hypothesis comparing the effect of civil unrest on child health care in the two independent populations ($n_i = 2(((Z_{1\_\alpha/2} + Z_{1\_\beta})/ES)^2)$). The hypotheses of interest were:

$H_0: P_1 = P_2$ versus $H_1: P_1 \neq P_2$. Where $p_1$ and $p_2$ are the proportions in the two populations (Pre- and into the civil unrest). With 95% confidence level, power of the test = 90% and $p_2$ = 0.5, and an assumption of a 25% increase in the proportion of children presenting with acute medical complications at a presentation during the civil unrest ($P_1$), the sample size was calculated to be 311 for each of the pre-and into the civil unrest periods.

A simple random sampling of registered medical record numbers (MRN) of children admitted to the pediatric ward over the study periods was implemented separately using Microsoft Excel.

## Exclusion criteria

Children whose MRN was missing or not addressing appropriate medical record files.

Children with incomplete Medical record file documentation (Fig 1).

## Definition, diagnosis, and classification

**Civil unrest.**  Is a collective term that includes limited political violence (acts of 'terrorism', individual assassinations), sporadic violent collective action (riots), or nonviolent and mildly violent collective action (protests) that causes tension or dissatisfaction over political, economic or social changes [8].

**Acute medical complications.**  Are unfavorable results of a disease, health condition, or treatment that may adversely affect the prognosis and outcome of a disease. They generally can be a worsening in the severity of the disease or the development of new signs, symptoms, or pathological changes that may become widespread throughout the body and affect other organ systems. Complications may lead to the development of new diseases resulting from previously existing diseases. Complications may also arise as a result of the different types of treatments provided [23]. Delay in diagnosis and treatment increases the risk of complications [24].

## Statistical analysis

This study employs the PSM-DID method to quantitatively examine whether and to what extent the civil unrest increases the acute medical complications at presentation to our

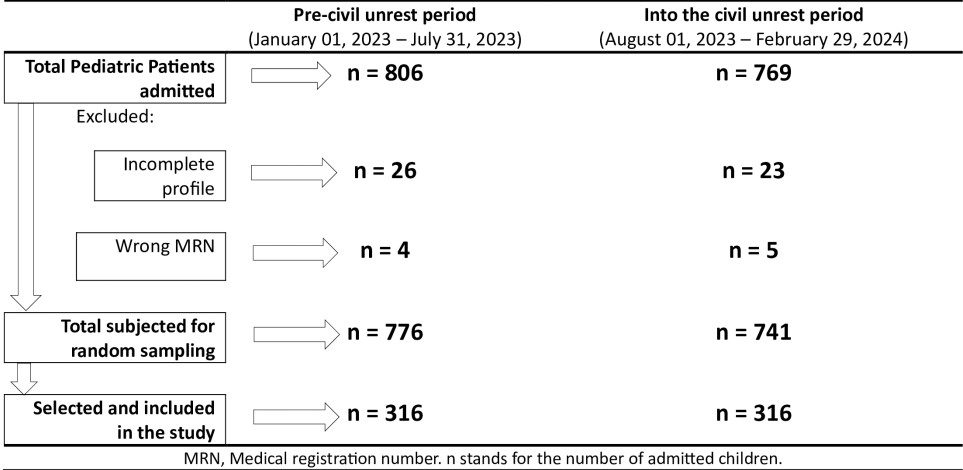

**Fig 1.  Flow chart of sampling processes for the observations selection, pre- and into the civil unrest periods.**

hospital on children from rural residency. DID method is widely acknowledged as the best method to study quasi-natural experiments or to evaluate the influences of economic crises, political turmoil, health interventions or policy implementation [25,26]. In our study, the DID recognizes rural residency as the treatment group, while the others are regarded as the control group. Differences for each group in acute medical complication proportion at presentation when compared before and after the civil unrest are then calculated. The difference in the above-mentioned differences thus represents the net effect of the "civil unrest" on children from rural residency manifested by acute medical complications at presentation to our hospital (Table 1).

Common trend assumption is one of the central assumptions of the DID method, which requires that the treatment group and control group should have parallel variation trends in terms of acute medical complications at presentation to our hospital before the civil unrest. This diagnostics for parallel trends is illustrated either graphically or statistically supported by the Parallel-trends test and Granger causality test. Parallel-trends test performs a test of whether the linear trends in the outcome variable are parallel between control and treatment groups during the pretreatment period. The Prob > F value represents the p-value associated with this test. If the Prob > F value is less than a significance level (commonly set at 0.05), we reject the null hypothesis and conclude that parallel trends are violated. Conversely, if the Prob > F value is greater than the significance level, we fail to reject the null hypothesis, suggesting that parallel trends are not significantly violated. The Granger causality test performs a test of whether treatment effects can be observed in anticipation of the treatment. In the Granger causality test, the Prob > F value represents the p-value associated with the F-statistic. A small Prob > F (typically less than 0.05) suggests evidence against the null hypothesis, supporting Granger causality.

The DID method requires a completely random selection between the treatment group and control group; otherwise, the result can be largely biased. However, rural residents could have been selected due to certain reasons, such as distance from the hospital. To relieve the endogenous problems caused by the selection bias, we adopt the Propensity Score Matching (PSM) method before applying DID, to ensure an accurate estimation of the impact of the civil unrest. PSM calculates the probability of a child being selected from a rural area. This study uses the Probit model to estimate the PSM. The results of PSM will enable us to identify the children in the control group that have the most similar possibility with the treated children to be selected as a child from rural residency. After excluding children who fail the matching process, there should be no significant difference in the matching variables between the treatment group and control group children. PSM thus can serve to relieve the selection bias and ensure the randomness of children selected as those coming from rural residences, improving the accuracy of the DID estimation. Finally, we performed impact estimation using the difference–in–differences (DID) method.

**Table 1.  Meaning of the DID method.**

| | Before August 2023 | After August 2023 | Difference |
|---|---|---|---|
| Treatment group (rural residency) | $\beta_0 + \beta_1$ [1] | $\beta_0 + \beta_1 + \beta_2 + \beta_3$ | $\Delta Y_t = \beta_2 + \beta_3$ |
| Control group (Urban residency) | $\beta_0$ [2] | $\beta_0 + \beta_2$ [3] | $\Delta Y_0 = \beta_2$ |
| DID | | | $\Delta\Delta Y = \beta_3$ [4] |

[1]Is the difference between the control and treatment group before the treatment.

[2]Is the average outcome of the control group before the treatment.

[3]Is the difference between the average outcome of the control group before and after the treatment.

[4]Difference-in-difference estimator. Does the treatment have an impact?

### Variables and data

**Key explanatory variables.** Dummy treatment is a dummy variable indicating whether a residency is selected as a rural residency; for rural residency, the value of dummy treatment is set as 1, otherwise it is set as 0. The dummy post is a month dummy variable representing the month when the civil unrest started; the value of the dummy post is set as 1 for months into the civil unrest (in and after August 01, 2023) and dummy post = 0 for months prior to the civil unrest (before August 01, 2023). The interaction term "Dummy treatment * dummy post" is the key explanatory variable (DiD) in this study, with its coefficient showing the effect of civil unrest on the proportion of acute medical complications at presentation to our hospital. According to our hypothesis, the coefficient should be significantly positive, demonstrating that children from rural residences are more affected by acute medical complications at presentation after the civil unrest.

**Control variables.** Matching variables were selected based on their theoretical association with the development of acute medical complications in children presenting with illness. In consideration of the literature and data availability, our study controls the following variables: age of children, sex, anthropometric variables, distance from the hospital, admission diagnosis, and vaccination status.

In the end, 632 admitted children (316 in each pre- and into the civil unrest period) enter into the PSM-DID analysis, among which 345 are in the treatment group and 287 are in the control group. STATA version 17 was used for analysis and SPSS version 27 was used for data cleaning. The two-sided statistical significance level was set to be 0.05.

## Results

### Characteristics of study samples

In seven months, prior to the civil unrest period, 806 children were admitted to the pediatric ward. In another seven months, into the civil unrest period, 769 children were admitted to the pediatric ward. Three hundred sixteen (316) from each time period were included in the study. After the selection, each participant, both in the pre-and into the civil unrest periods was assigned to a group based on their residency location (treatment vs. control). 56.6% were males. The median age (IQR) was 22 (7, 77.3) months (Table 2).

### The trend of acute medical complications over the study period

The trend of acute medical complications at presentation has increased in both children coming from rural and urban areas during the civil unrest period (Fig 2). Ninety five per cent of all children with acute Gastro-intestinal complications were presenting during the civil unrest period followed by acute musculoskeletal complications (87%) and acute complications presenting as fluid and electrolyte disturbance (85%) respectively (S1 Table).

### Propensity score matching (PSM)

To eliminate the selection bias and ensure the accuracy of DID analysis; we first conducted a PSM on the 632 admitted children, including 345 treated and 287 control children. The matching variables are sex, age category, distance to the hospital, wasting, stunting, vaccination status, and admission diagnosis. PSM was conducted on 7 covariates. In the unmatched sample, significant differences between groups were found for two covariates including distance from the hospital and age category of children. The treatment effects before matching (Unmatched) and after matching (ATT), which are reflected as the risk difference between treated and control units for developing acute medical complications, are 0.164 and 0.241

**Table 2. Characteristics of study samples, pre- and into the civil unrest period.**

| Variables | Pre-civil unrest | | | | Into the civil unrest | | | |
|---|---|---|---|---|---|---|---|---|
| | Control (Urban) | | Treatment (Rural) | | Control (Urban) | | Treatment (Rural) | |
| | Number | Percent | Number | Percent | Number | percent | Number | Percent |
| **Distance to hospital** | | | | | | | | |
| Less than 100 kms | 104 | 32.9% | 74 | 23.4% | 109 | 34.5% | 85 | 26.9% |
| Greater or equal to 100 kms | 42 | 13.3% | 96 | 30.4% | 32 | 10.1% | 90 | 28.5% |
| **Child's age category** | | | | | | | | |
| 1 - <12 months | 53 | 16.8% | 54 | 17.1% | 64 | 20.3% | 56 | 17.7% |
| ≥12 months - <24 months | 24 | 7.6% | 24 | 7.60% | 23 | 7.3% | 25 | 7.9% |
| ≥24 months - <72 months | 31 | 9.8% | 35 | 11% | 26 | 8.2% | 40 | 12.7% |
| ≥72 months - <144 months | 28 | 8.9% | 34 | 10.8% | 18 | 5.7% | 36 | 11.4% |
| ≥14 4months | 10 | 3.1% | 23 | 7.3% | 10 | 3.1% | 18 | 5.7% |
| **Sex of child** | | | | | | | | |
| Male | 81 | 25.6% | 91 | 28.8% | 82 | 25.9% | 104 | 32.9% |
| Female | 65 | 20.6% | 79 | 25% | 59 | 18.7% | 71 | 22.5% |
| **Wasting** | | | | | | | | |
| No | 108 | 34.2% | 121 | 38.3% | 116 | 36.7% | 125 | 39.6% |
| Yes | 38 | 12% | 49 | 15.5% | 25 | 7.9% | 50 | 15.8% |
| **Stunting** | | | | | | | | |
| No | 119 | 37.7% | 142 | 44.9% | 129 | 40.8% | 140 | 44.3% |
| Yes | 27 | 8.5% | 28 | 8.9% | 12 | 3.8% | 35 | 11.1% |
| **Vaccination status** | | | | | | | | |
| Not vaccinated | 14 | 4.4% | 28 | 8.9% | 14 | 4.4% | 21 | 6.7% |
| Vaccinated | 132 | 41.8% | 142 | 44.9% | 127 | 40.2% | 154 | 48.7% |
| **Admission diagnosis category** | | | | | | | | |
| Surgical diagnosis | 44 | 13.9% | 74 | 23.4% | 66 | 20.9% | 68 | 21.5% |
| Medical diagnosis | 102 | 32.3% | 96 | 30.4% | 75 | 23.7% | 107 | 33.9% |
| **Acute medical complication at presentation** | | | | | | | | |
| No | 126 | 39.9% | 143 | 45.3% | 94 | 29.7% | 65 | 20.6% |
| Yes | 20 | 6.3% | 27 | 8.5% | 47 | 14.9% | 110 | 34.8% |
| **Total** | **146** | **46.2%** | **170** | **53.8%** | **141** | **44.6%** | **175** | **55.4%** |

respectively. This tells us that had the treated units not been treated, their estimated risk of developing acute medical complications would be 0.241 lower than observed (Table 3).

The propensity score distribution for treated and control groups before and after matching suggests the control group meets the common support assumption after matching (Fig 3).

Propensity Score Matching is applied to estimate whether children in the control group have no difference from the treated group in terms of the matching variables selected. The estimated bias of all variables after matching is less than 20%. The t-test results for all variables are non-significant after matching. This suggests that almost all observable covariates are sufficiently balanced by the matching between the treatment and control groups. The initial differences in the two groups are reduced considerably and have become statistically insignificant at 5%. Hereafter, the samples are suitable for DID analysis (Table 4).

## Parallel trend test

To test the parallel trend assumption, the proportion of acute medical complications at presentation to our hospital of both groups (Fig 4) illustrates the trends of the average value of acute

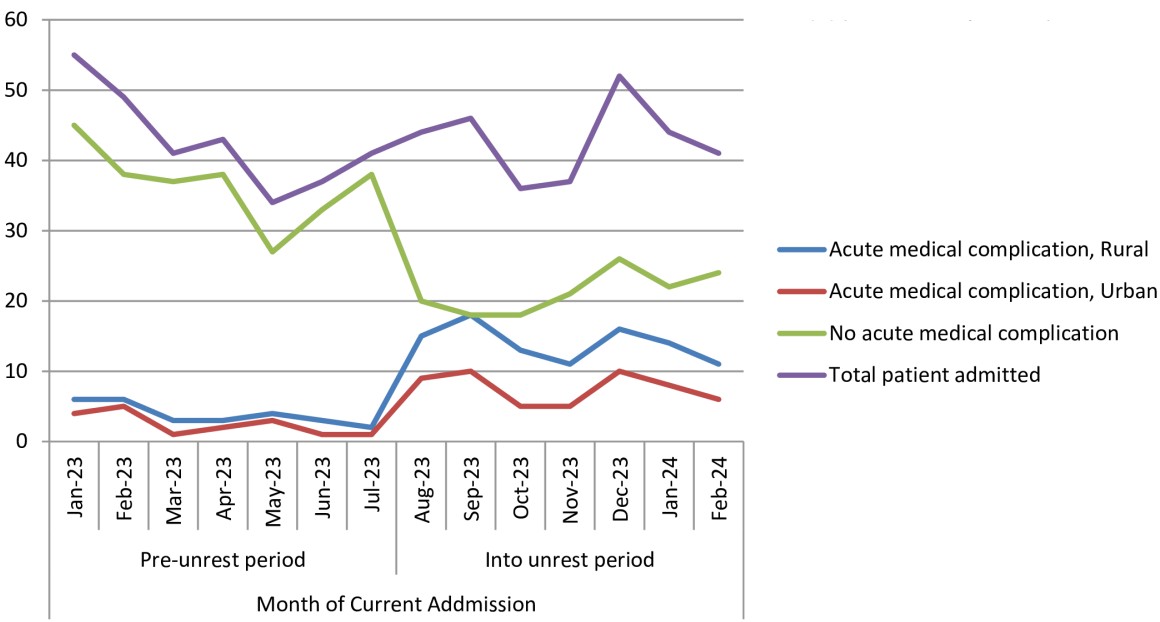

**Fig 2. Trends of acute medical complications at presentation in children from urban and rural residency prior and into the civil unrest periods.**

**Table 3. Propensity Score Matching on the covariates. ATT, the average treatment effect on the treated.**

| Probit regression | | | Number of obs = 632 | | | |
|---|---|---|---|---|---|---|
| Residency | Coefficient | Std. err. | Z | P>\|z\| | {95% conf. interval} | |
| SEX | -0.066961 | 0.1056735 | -0.63 | 0.526 | -0.2740772 | 0.1401552 |
| Age Category | 0.1109609 | 0.0380635 | 2.92 | 0.004 | 0.0363578 | 0.185564 |
| DistancefromHospital | 0.7374586 | 0.1072628 | 6.88 | 0.000 | 0.5272273 | 0.9476898 |
| AdmissionDiagnosis | 0.0083692 | 0.1107728 | 0.08 | 0.940 | -0.2087414 | 0.2254799 |
| Wasting | 0.1467649 | 0.1630015 | 0.90 | 0.368 | -0.1727123 | 0.466242 |
| Stunting | 0.084817 | 0.1928214 | 0.44 | 0.660 | -0.293106 | 0.46274 |
| Vaccination | -0.2369789 | 0.164973 | -1.44 | 0.151 | -0.56032 | 0.0863623 |
| _cons | -0.205408 | 0.229009 | -0.90 | 0.370 | -0.6542575 | 0.2434415 |
| Variable | Sample | Treated | Controls | Difference | S.E | T-stat |
| Acute medical complications | Unmatched | 0.397101449 | 0.233449477 | 0.163651972 | 0.036839933 | 4.44 |
| | ATT | 0.398255814 | 0.156976744 | **0.24127907** | 0.063038757 | 3.83 |

S.E. does not take into account that the propensity score is estimated.

medical complications at presentation, from January 01, 2023, to February 29, 2024. The red line highlights the month when the civil unrest had started. Judging from the trend lines, the treatment group admitted children have a rather similar variation trend with the control group admitted children before August 01, 2023. And this supports the common trend assumption. Moreover, after entering into the civil unrest period, the treatment group showed a larger increment in acute medical complications at presentation, compared with the control group which experienced little fluctuations after August 01, 2023. This generates a preliminary finding that civil unrest can increase acute medical complications at the presentation of children with illnesses to our hospital.

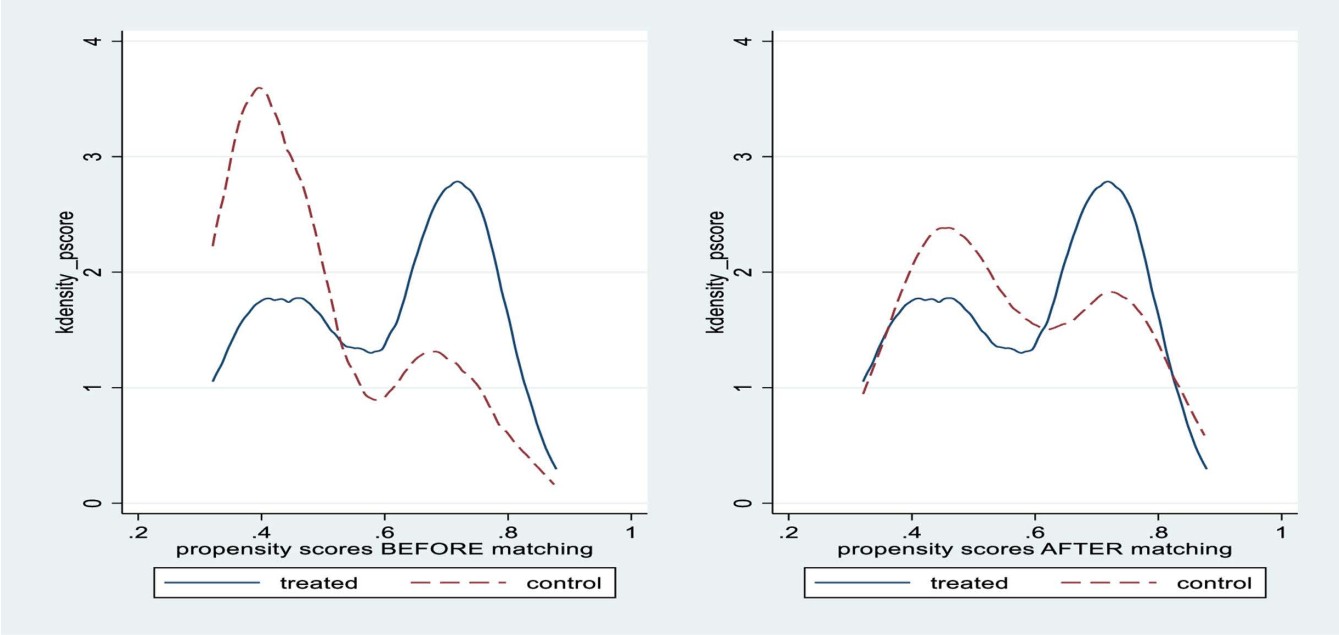

**Fig 3. Propensity score distribution before and after matching.**

**Table 4. Balance test for variables after Propensity score matching/PSM/.**

| Variable | Mean | | %Bias | t-test | |
|---|---|---|---|---|---|
| | Treated | Control | | t | p> \|t\| |
| SEX of the child | 1.436 | 1.4913 | -11.1 | -1.45 | 0.147 |
| Age Category | 2.6541 | 2.6977 | -3.2 | -0.41 | 0.683 |
| Distance to Hospital | 0.53779 | 0.55523 | -3.7 | -0.46 | 0.646 |
| Admission Diagnosis | 0.59012 | 0.61047 | -4.2 | -0.54 | 0.587 |
| Wasting | 0.28488 | 0.25291 | 7.4 | 0.95 | 0.345 |
| Stunting | 0.18314 | 0.19186 | -2.4 | -0.29 | 0.770 |
| Vaccination | 0.86047 | 0.90698 | -14.3 | -1.91 | 0.057 |

The graph indicates that the parallel-trends assumption is acceptable. Prior to the civil unrest, acute medical complications at presentation to the hospital from treated (rural residency) and control (urban residency) followed a parallel path. This graphical diagnostics for parallel trends is statistically supported by Parallel-trends (pretreatment time period) and Granger causality tests. The Prob > F value for parallel-trends test is greater than the significance level (0.422). Hence, we fail to reject the null hypothesis, suggesting that parallel trends are not significantly violated. Similarly, in the Granger causality test, the Prob > F value is 0.1385, which is greater than p-value 0.05, suggesting evidence in favor of the null hypothesis "no Granger causality" (Tables 5 and 6).

## Main regression results

The regression result of acute medical complications at presentation to our hospital shows the average treatment effect on the treated (ATET) to be 0.24. This is a 0.24-point increase in acute medical complications in children coming from rural area (treated) relative to those

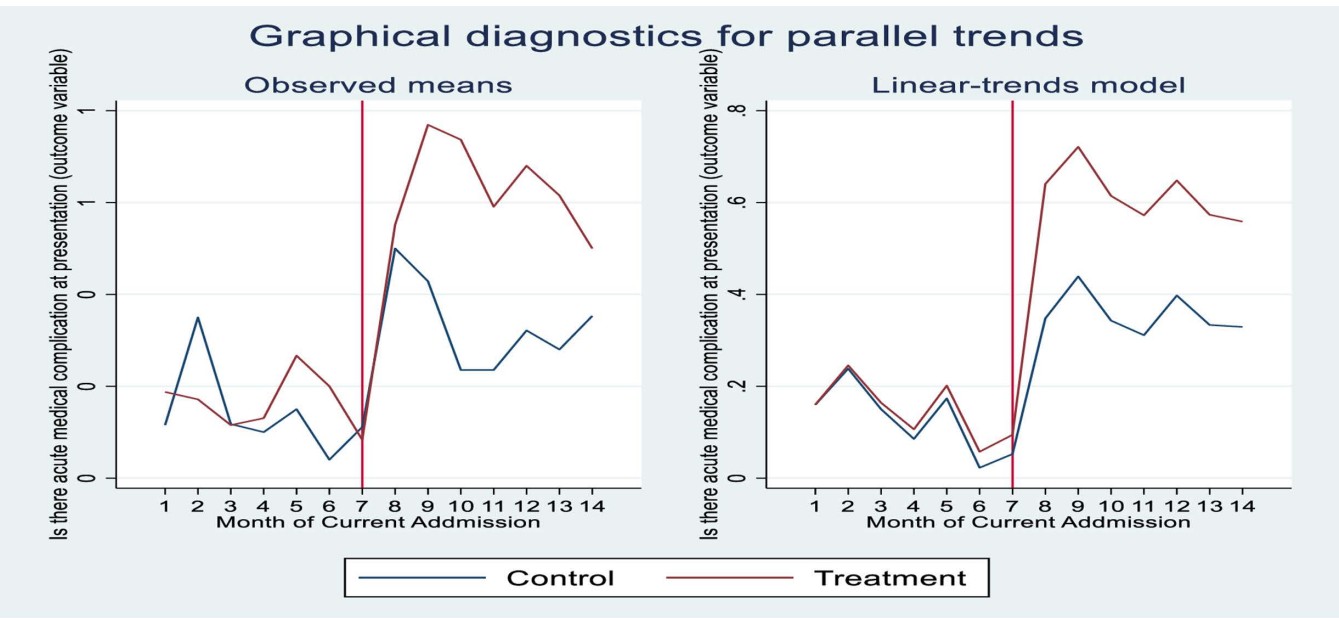

**Fig 4. Parallel trends of acute medical complications.** Red, Treatment (Rural). Blue, control (Urban).

**Table 5. Parallel-trends test (pre-treatment time period).**

| |
|---|
| . estat ptrends |
| Parallel-trends test (pretreatment time period) |
| H0: Linear trends are parallel |
| F($1, 1$) = 1.64 |
| Prob > F = 0.4220 |

**Table 6. Granger causality test.**

| |
|---|
| . estat granger |
| Granger causality test |
| H0: No effect in anticipation of treatment |
| F($1, 1$) = 20.48 |
| Prob > F = 0.1385 |

from urban areas (control). This increment is statistically significant at a p-value of 0.009 (Table 7).

## Discussion

Our study revealed that acute medical complications at admission have increased significantly in children coming from rural areas compared to their counterparts from urban areas during the civil unrest period, as depicted by the DID value=0.241 with p value=0.009. Up to our knowledge, no research clearly shows the impact of civil unrest on child health care revealing nuanced statistical results on conflict's effect on children's health in such a way that enables to compare "children living in the same region of a same country" with different exposure to

**Table 7. Regression results of acute medical complications at presentation at residency level.**

. didregress (AcuteMedicalComplication DistancefromHospital AGECATEGORY SEX> Wasting Stunting Vaccination AdmissionDiagnosis) (DiD), group(Residency)> time (MonthOFcurrentAdmission)

| Number of groups and treatment time | | | | | | | | |
|---|---|---|---|---|---|---|---|---|
| Time variable: | | MonthOFcurrentAdmission | | | | | | |
| Control: | | DiD = 0 | | | | | | |
| Treatment: | | DiD = **1** | | | | | | |
| | | Control | Treatment | | | | | |
| Group | Residency | 1 | 1 | | | | | |
| Time | Minimum | 1 | 8 | | | | | |
| | Maximum | 1 | 8 | | | | | |
| Difference-in-differences regression | | | | | Number of obs = 632 | | | |
| Data type: Repeated cross-sectional | | | | | | | | |
| (Std. err. adjusted for 2 clusters in Residency) | | | | | | | | |
| Acute medical Complication | | | Robust | | | | | |
| | | Coefficient | Std. err. | T | P>\|t\| | {95% conf.interval} | | |
| ATET | DiD (1 vs 0) | .2410636 | .0033277 | 72.44 | 0.009 | .1987807 | | .2833465 |

ATET estimate adjusted for covariates, group effects, and time effects.

armed conflict over time and place of residency. Most of the researches done on this matter are focusing more on demonstrating the prevalence and diseases profiles of children in conflict-hit areas [15,27,28].

It is already researched and documented that there are clear, negative health effects in countries affected by civil unrest [29,30]. In addition, conflicts can also distract governments from allocating sufficient attention and resources to the health care facilities in the civil unrest-hit areas which in turn affects the health care and outcome of sick children who already arrived at the health facilities for their illness in time. Hence; children are developing complications, that otherwise be treated easily without the due development of acute complications.

Rural and urban areas have different health challenges and resources for their residents in that urban areas usually have a better local supply, adequate density of public transport, better access to education, a rather health-promoting environment, and a high density of health care facilities that all help increase children's healthcare utilization opportunity compared to their counterpart rural area children[31]. In a conflict-hit area, these all privileges are missing and children from rural area are going to miss it more[32].

The acute medical complications developed by children presenting to our hospital are higher in children coming from rural areas compared to those coming from urban areas. Much of the differences in healthcare outcomes between rural and urban residents can be attributed to the differential rates of primary healthcare utilization and resource allocation across the population subgroups [33]. In addition, this can be partly explained by low awareness levels, geographical and transportation-related barriers, decreased availability and accessibility of primary care and specialty care, inadequate health education, interruption of health insurance coverage, and socioeconomic determinants, especially poverty and substandard housing conditions [34,35].

The proportion of children presenting with acute medical complications is higher in those with gastrointestinal, followed by musculoskeletal and fluid and electrolyte abnormality complications respectively. Had it been for non-conflict areas, the underlying disease conditions of these complications were to be addressed at the front-line health facilities with less intensive and cheaper expenses with the available human resource for health.

In a fragile health situation due to civil unrest like in our case, timely detection, intervention, and referral of diseased children is almost unthinkable and are at a higher risk of developing the deadly complications of the underlying illnesses. Hence, there shall be a way designed to easily access the health service for those sick children from rural areas so that they will not be complicated anymore.

Therefore, stakeholders working at regional, national, and international levels should increase efforts to implement a strategy to promote facilitated access to health care at appropriate times in areas affected by the ongoing civil unrest—but the most critical intervention is to end and prevent war.

Our study provides useful insights into the outcomes of child health medical care in the setting of civil unrest. Because the study was done at one tertiary facility, the results may not be generalizable to other conflict settings. The study focuses on acute medical complications and doesn't examine the mental health outcome as a complication, which is one of the children's critical issues in the setting of civil unrest. Our study is a retrospective one which may not be able to capture all covariates from the medical records and the PSM–DID model should contain all covariates that may influence the intervention effect before and after matching. Hence, unobservable covariates may cause different trends between the treatment group and the control group, and such an instance may have led to biased results.

## Conclusion

Our study has concluded that civil unrest has an immediate impact on child health care evidenced by an increased proportion of acute medical complications at presentation. It affects more children coming from rural areas compared to those from urban communities.

## Supporting information

**S1 File. Impact of unrest raw data 1.**
(ZIP)

**S1 Table. Types of acute medical complications.**
(DOCX)

AcknowledgmentWe thank the Pediatric and Child health residents and Clinical nurses working at the pediatric ward of Bahir Dar University Tibebe-Ghion Specialized Teaching Hospital for their relentless support in collecting the necessary registries and patient medical files. Our thanks also go to the administration of the hospital for allowing us to collect the entire necessary document.

## Author contributions

**Conceptualization:** Tesfaye Taye Gelaw.

**Data curation:** Tesfaye Taye Gelaw, Gasha Amsalu Tadesse.

**Formal analysis:** Tesfaye Taye Gelaw, Gebeyaw Lulie Adamu.

**Investigation:** Tesfaye Taye Gelaw, Kassahun Gedefie Hailu.

**Methodology:** Tesfaye Taye Gelaw.

**Project administration:** Tesfaye Taye Gelaw, Mensur Azeze Getahun.

**Resources:** Tesfaye Taye Gelaw.

**Software:** Tesfaye Taye Gelaw.

**Supervision:** Tesfaye Taye Gelaw, Assefa Mitiku Bayih, Mastewal Addisu Legesse.

**Validation:** Tesfaye Taye Gelaw.

**Visualization:** Tesfaye Taye Gelaw.

**Writing – original draft:** Tesfaye Taye Gelaw.

**Writing – review & editing:** Tesfaye Taye Gelaw, Mensur Azeze Getahun, Kassahun Gedefie Hailu, Assefa Mitiku Bayih, Gasha Amsalu Tadesse, Mastewal Addisu Legesse, Gebeyaw Lulie Adamu.

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
