## [Decision Letter · Decision Letter 0]

4 Nov 2024

PONE-D-24-26693The Impact of civil unrest on child health care: evidenced by acute medical complications at presentation – a retrospective comparative study.PLOS ONE

Dear Dr. Gelaw,

Thank you for submitting your manuscript to PLOS ONE. After careful consideration, we feel that it has merit but does not fully meet PLOS ONE’s publication criteria as it currently stands. Therefore, we invite you to submit a revised version of the manuscript that addresses the points raised during the review process.

We look forward to receiving your revised manuscript.

Kind regards,

Sk Md Mamunur Rahman Malik

Academic Editor

PLOS ONE

Journal Requirements:

https://unesdoc.unesco.org/ark:/48223/pf0000190712

https://pubmed.ncbi.nlm.nih.gov/30621338/

In your revision ensure you cite all your sources (including your own works), and quote or rephrase any duplicated text outside the methods section. Further consideration is dependent on these concerns being addressed.

Reviewers' comments:

Reviewer's Responses to Questions

**Comments to the Author**

1. Is the manuscript technically sound, and do the data support the conclusions?

Reviewer #1: Partly

Reviewer #2: Yes

2. Has the statistical analysis been performed appropriately and rigorously? 

Reviewer #1: I Don't Know

Reviewer #2: Yes

3. Have the authors made all data underlying the findings in their manuscript fully available?

Reviewer #1: Yes

Reviewer #2: Yes

4. Is the manuscript presented in an intelligible fashion and written in standard English?

Reviewer #1: Yes

Reviewer #2: Yes

5. Review Comments to the Author

Reviewer #1: Thank you for your manuscript. Please see the attached document for detailed comments and suggestions.

Reviewer #2: The authors have presented a paper on the topic ‘The Impact of civil unrest on child health care: evidenced by acute medical complications at presentation – a retrospective comparative study.’ Generally, the paper is clearly written and makes good contribution to research on children’s wellbeing. However, minor issues have been identified and it is recommended that these be addressed before publication of the manuscript. Please see below comments:

The new sentence on the second line under the Introduction should start with a capital C, that is ‘Children…’

In-text citations throughout the paper are written outside the sentences after the punctuation mark- full stop is written. It will be good to bring citations before the punctuation mark to avoid adding citations to the beginning of new sentences.

Referring to these statements ‘Data were collected by trained nurses and pediatric residents. Authors had no access to identify participants of the study’, it appears the authors used secondary data from medical records taken by others. It will be helpful for the authors to clarify the type of data used in the study, whether the data is considered primary or secondary data to guide future studies and whether any of the authors were involved in the data collection.

In text reference to tables and figures should not be placed after the full stop but before.

Generally, I believe that the discussion should be improved by providing more information and explanations to the findings. As the findings are very technical, for easy understanding by readers, the discussion is required to explain the findings and implications more clearly but I feel that this is lacking in the discussion section. It is very brief, some sentences appear vague and does not highlight more on all the key findings for example, what types of complications were identified. Indeed, there is a general presentation of trends showing an increase in complications and rural residents been affected more, but as to what types of complications they experienced more than the urban residents or in what ways are they more affected based on the findings are not explained.

If there is a lack of awareness/data of the types of acute complications that the patients experienced, then this can be considered as limitation of the study as it will make a big difference in the study if there is a presentation of some of the acute complications identified to affect the groups differently. This will provide guidance for targeted intervention strategies.

6. PLOS authors have the option to publish the peer review history of their article (what does this mean? ). If published, this will include your full peer review and any attached files.

**Do you want your identity to be public for this peer review?** For information about this choice, including consent withdrawal, please see our Privacy Policy .

Reviewer #1: **Yes: ** Beth A Tippett Barr

Reviewer #2: No

---

## [Author Response · Author response to Decision Letter 1]

13 Nov 2024

Response to academic editor and reviewers (Rebuttal letter):

Journal: PLOS ONE

Manuscript number: PONE-D-24-26693

Title: The Impact of civil unrest on child health care: evidenced by acute medical complications at presentation – a retrospective comparative study.

Dear academic editor Sk Md Mamunur Rahman Malik and reviewers,

Thank you very much for your nourishing comments and feedbacks. All the comments are constructive and lessons for us.

Here below are our point-by-point responses and corrections for the constructive feedbacks provided.

Response to Academic Editor Feedbacks:

a. Response: comment well accepted and revised as per PLOS ONE’s style.

a. Response: comment well accepted. The paragraphs are revised and rephrased.

Response to review comments of Reviewer #1:

1. Comment 1: Overall: Do a careful reading for grammatical corrections.

a. Response: comment accepted and grammatical corrections done throughout the document.

2. Comment 2: Background/Introduction

a. Information about what happens to healthcare providers during conflict in your country context :

i. Response: comment well accepted and information regarding healthcare providers is included in the background/introduction part.

b. Final sentence: ……. The need to inform the government …..

i. Response: Comment well accepted and included.

3. Comment 3: Methods

a. Study design – you have included a lot of information in this section about the statistical analysis, which belongs in the statistical analysis section. For the study design section, it’s okay to be brief and define this as a retrospective medical record review.

i. Response: comment well accepted and edited accordingly.

b. Study design – your ethics statement is in this section, but usually would have its own section or be shown separately in a section after the manuscript. Please review journal requirements and include as advised by the journal.

i. Response: comment well accepted and shown separately.

c. Definition – acute medical complications: There’s nothing in the definition to indicate this can be a result of delayed care, but I believe this is the premise of your paper; the care is delayed and as a result, children are more likely to have acute medical complications? If correct, please refine your definition to include a reference to delayed care.

i. Response: Comment accepted and refined.

d. Sample size – I don’t believe you need the full equations included. You can make this section far more concise by using wording only. See other journal articles for how others have done this.

i. Response: comment well accepted and made concise by using wording.

e. Table 1 – please provide table footnotes on what Beta represents.

i. Response: comment well accepted and table footnote provided.

4. Comment 4: Results:

a. Table 2. Please remove color stripes from the table. Please add column percentages so that it’s easier to ‘see’ the data distribution between groups

i. Response: comment accepted and color stripes are removed from the table.

ii. Response: comment accepted and column for percentages is added.

b. “table 1” in STATA (?) output – please remove this and create “Table 3”. In the narrative please remove definitions and just describe the findings objectively.

i. Response: comment accepted. The output table is removed and Table 3 created. This created table 3 highlights the treatment effect between covariates before and after matching. Definition is removed and findings are described.

c. “table 2” in STATA (?) output – please remove this and create “Table 4” – if possible, consider merging these two STATA tables into a single table for the results. Think how best to share your findings in a way that’s both understandable and concise

i. Response: STATA output removed and “Table 4” created. This created table 4 the balance test for variables after PSM. Actually we used the font similar to STATA output formats and it seems a copy. Merging may limit someone’s understanding of our work and as the column and row numbers are different for the two tables, it might lead us miss the content. Hence, with due respect, we left it as it is (two tables).

d. “Interpretation” – remove this statement and put everything into narrative that describes your findings.

i. Response: comment accepted.

e. “table 5” in STATA output – remove this as well and describe in the narrative.

i. Response: comment accepted and description included in the narration.

5. Comment 5: Discussion

a. You need to describe a bit more about how the other research studies have documented negative health effects if not at the individual level – what did they look at that was not as rigorous as your study?

i. Response: comment well accepted and the negative health effects of conflict in other studies is narrated as “Up to our knowledge, no research clearly shows the impact of civil unrest on child health care revealing nuanced statistical results on conflict’s effect on children’s health in such a way that enables to compare “children living in the same region of a same country” with different exposure to armed conflict over time and place of residency.” Most of the researches done on this matter are focusing more on demonstrating the prevalence and diseases profiles of children in conflict-hit areas.

b. In the third paragraph which starts “the acute medical complications” – please describe a bit more about the differences between rural and urban before and during conflict – it doesn’t come out clearly here

i. Response: comment well accepted and description on the difference is included.

6. Comment 6: Limitations

a. Remove the word ‘holistically’ – be more concise; you can just say that because the study was done at one tertiary facility, the results may not be generalizable to other conflict settings

i. Response: comment well accepted and re-iterated as recommended.

7. Comment 7: Conclusion, Recommendation, Implications

a. There are too many sections here. You should have your “Discussion” and “Conclusion” sections. Your implications all need to be incorporated into your discussion section, connected to the key findings that you shared in the results section. Aim to have about 3-4 main points (1 per paragraph) and work the implications into each paragraph so you have a coherent discussion that flows well. Your limitations section should be the final paragraph in your discussion.

i. Response: comment well accepted and re-written as recommended.

Response to review comments of Reviewer #2:

1. Comment 1: The new sentence on the second line under the Introduction should start with a capital C, that is ‘Children …’

a. Response: comment accepted and edited

2. Comment 2: In-text citations throughout the paper are written outside the sentences after the punctuation mark- full stop is written. It will be good to bring citations before the punctuation mark to avoid adding citations to the beginning of new sentences.

a. Response: comment well accepted and edited.

3. Comment 3: Referring to these statements ‘Data were collected by trained nurses and pediatric residents. Authors had no access to identify participants of the study’, it appears the authors used secondary data from medical records taken by others. It will be helpful for the authors to clarify the type of data used in the study, whether the data is considered primary or secondary data to guide future studies and whether any of the authors were involved in the data collection.

a. Response: Comment well accepted and re-phrased as ‘Data were collected by properly trained nurses and pediatric residents retrospectively using structured questionnaire from medical records of children admitted to pediatric ward ‘. Our data is primary data collected for this study from medical records of clients admitted to the hospital.

4. Comment 4: In text reference to tables and figures should not be placed after the full stop but before.

a. Response: comment well accepted and edited.

5. Comment 5: Discussion

a. Response: comment is well accepted. The type of complications are included as a supplement table and narrated in the result section as well as the discussion part.

Thank you for all your constructive comments,

Corresponding author

Dr. Tesfaye Taye Gelaw

---

## [Editor Report · Decision Letter 1]

27 Feb 2025

The Impact of civil unrest on child health care: evidenced by acute medical complications at presentation – a retrospective comparative study.

PONE-D-24-26693R1

Dear Dr. Gelaw,

We’re pleased to inform you that your manuscript has been judged scientifically suitable for publication and will be formally accepted for publication once it meets all outstanding technical requirements.

Kind regards,

Sk Md Mamunur Rahman Malik

Academic Editor

PLOS ONE
---

## [Editor Report · Acceptance letter]

PONE-D-24-26693R1

PLOS ONE

Dear Dr. Gelaw,

I'm pleased to inform you that your manuscript has been deemed suitable for publication in PLOS ONE. Congratulations! Your manuscript is now being handed over to our production team.

Kind regards,

on behalf of

Dr. Sk Md Mamunur Rahman Malik

Academic Editor

PLOS ONE
